# The Performance of Learners’ Strategic Flexibility and Its Relationship with External Factors and Cognitive Flexibility: A Survey of High School Mathematics in China

**DOI:** 10.3390/bs15111440

**Published:** 2025-10-23

**Authors:** Xinyuan Yang, Kui Feng, Yong Zhang, Bin Xiong

**Affiliations:** 1School of Mathematical Sciences, East China Normal University, Shanghai 200241, China; 52170601003@stu.ecnu.edu.cn; 2Shanghai Key Laboratory of Pure Mathematics and Mathematical Practice, Shanghai 200241, China; 3Pu’er No. 1 Middle School, Pu’er 665000, China; 4School of Mathematics, Yunan Normal University, Kunming 650500, China

**Keywords:** mathematical flexibility, actual flexibility, potential flexibility, cognitive flexibility, survey research

## Abstract

As a behavioral ability, flexibility plays an indispensable role in human learning activities. However, the analysis of flexibility in specific disciplines has not yet been fully explored. In response, through trigonometry of mathematics, this study investigated the strategic flexibility of high school level students, examining the influence of external factors such as gender and class on flexibility, and exploring the relationship between cognitive and strategic flexibility. Based on the four-stage flexibility test and the cognitive flexibility questionnaire survey of 237 11th-grade students in China, the current research yielded the following findings: (1) There is a positive correlation between potential strategic flexibility and actual strategic flexibility, and the actual strategic flexibility level of students is higher than that of potential strategic flexibility. (2) Gender and class have no significant relationship with strategic flexibility, but different subject combinations have a certain impact on flexibility. (3) Cognitive flexibility has a positive effect on both potential strategic flexibility and actual strategic flexibility. These findings have provided some research basis for understanding students’ external performance and adjusting teachers’ teaching behaviors, proposing a certain adjustment direction for the focus of teaching, students’ learning content, and classroom teaching methods.

## 1. Introduction

Flexibility, as an adjustment ability made in response to changes in circumstances, plays a significant role in the learning process. It not only promotes the formation of adaptive behaviors among individuals but also enables multi-dimensional and flexible switching of perspectives, optimizing decision-making and problem-solving, and enhancing learning efficiency and problem-solving effectiveness. Furthermore, flexibility is also conducive to the emergence of innovative behaviors. By overcoming habitual activities in thinking and behavior, promoting the integration of subconscious information, it can break through the traditional behavioral framework and achieve behavioral innovation. As a core dimension of cognitive ability, flexibility holds multi-dimensional value in personal development and social adaptation, which makes relevant research on flexibility extremely necessary and of profound significance.

### 1.1. Mathematical Flexibility

In mathematics education, flexibility is a crucial learning goal that has been a significant focus in educational psychology research. Chinese scholars have primarily examined flexibility through the lens of mathematical thinking qualities. [16] ([16]), summarizing over 40 years of cognitive psychology studies, emphasized the importance of cultivating students’ thinking qualities to develop intelligence and enhance teaching quality. He described mathematical thinking flexibility as a reflection of the scientific level of rapid shifts in thinking direction, processes, and skills during mathematical tasks. [14] ([14]), in his research on mathematical abilities of primary and secondary students, identified mathematical thinking flexibility as a key component of mathematical aptitude, defining it as the ability to switch between mental operations and break free from conventional limitations. Recent years have seen scholars worldwide further exploring mathematical flexibility, including assessment methods, influencing factors, and instructional improvement strategies ([10]; [11]). Mathematical flexibility is commonly conceptualized as the utilization of various strategies in problem-solving, with both narrow and broad perspectives. The narrow view focuses solely on the knowledge and application of different strategies ([7]; [9]; [28]; [36]). In contrast, the broad view encompasses both strategy knowledge and use, as well as the ability to select the most suitable strategy for a given problem ([3]; [25]; [30]; [32]). Within the broad flexibility framework, three main aspects have been extensively researched: strategic flexibility (procedural flexibility), representational flexibility, and adaptive number knowledge ([22]). Representational flexibility refers to the capacity to employ and switch between various mathematical representations (such as graphs, tables, language, and symbols) across different problem contexts ([1]). Adaptive number knowledge, focusing on elementary arithmetic, involves choosing and applying the most appropriate strategies for specific arithmetic problems, requiring students to recognize relevant numerical features and their relationships within problems ([22]).

### 1.2. Strategic Flexibility

This research investigates the concept of strategic flexibility, also known as procedural flexibility, which has been interpreted in various ways. One interpretation considers it the capacity to adaptively employ multiple strategies when tackling specific mathematical problems or related challenges. Another view encompasses not only the range of strategies but also the selection and implementation of the most suitable approach. These perspectives primarily emphasize the procedural aspect of strategic flexibility. Alternative conceptual frameworks incorporate both procedural and conceptual elements. Scholars such as Star characterize strategic flexibility as “(a) knowledge of multiple strategies, and (b) the ability and inclination to choose the most appropriate strategy given a specific problem and a defined problem-solving goal” ([29]). In additional research (e.g., [30]), they break down the two components of flexibility into four subcategories: awareness of multiple strategies, understanding of optimal strategies, utilization of multiple strategies, and application of optimal strategies.

The definition of strategic flexibility adopted by researchers directly impacts its measurement and investigation. Studies on flexibility often explicitly require participants to employ multiple problem-solving methods (e.g., [3]; [34]). However, this requirement is implemented differently across studies. Some researchers examine the number of methods used for a single problem (within-problem flexibility), while others investigate the ability to switch strategies across multiple problems (across-problem flexibility). Certain studies incorporate both approaches. In task-based assessments, researchers pre-determine the optimal strategy based on problem characteristics and analyze participants’ selection of this strategy. For example, [8] ([8]) examined whether sixth-grade Dutch students used short-cut strategies for specific multi-digit arithmetic problems. Participants completed a paper-based task with problems designed to elicit particular short-cut strategies, such as quickly solving 316 + 178 + 284 by adding the first and third numbers first. Strategies were categorized as short-cut or not based on responses, assessing problem-solving flexibility.

Many researchers utilize a “choice/no-choice” design, testing participants under two conditions: a choice condition allowing free strategy selection, and a no-choice condition requiring a specific strategy for all problems. Researchers integrating procedural and conceptual aspects analyze actual strategy choices and explore understanding of alternative strategies and their optimal application. The “three-phase flexibility assessment” method proposed by [30] ([30]) involves solving mathematical problems in three stages. First, learners solve problems quickly, with their processes recorded. Second, they generate multiple solutions for each problem. Finally, they select the optimal strategy from those previously generated. This method has been successfully applied by [37] ([37]) and [33] ([33]) in research on students’ flexibility in solving linear equations.

The majority of studies on mathematical strategic flexibility have concentrated on areas like integer operations, estimation, ratios and proportions, rational numbers, and basic algebra, primarily involving middle and elementary school students from foreign countries. There is a scarcity of empirical research on strategic flexibility of high school students among Chinese scholars. The Curriculum Standards for Mathematics in Senior High Schools (2022 revision of the 2017 edition) emphasizes trigonometric identity transformation, a vital aspect of trigonometric functions, recommending the use of diverse methods to derive basic formulas ([24]). Due to their skillfulness and multiple solutions, in the mathematics content of high school, trigonometric identity transformations are ideal for evaluating students’ mathematical thinking flexibility. However, the question of how Chinese high school students demonstrate strategic flexibility when tackling trigonometric identity transformation problems remains empirically unexplored.

### 1.3. Cognitive Flexibility

Cognitive flexibility is also a key area in child psychological development research. Martin et al. proposed the widely accepted theory of cognitive flexibility and created an effective measurement questionnaire ([20]; [18]). They conceptualized cognitive flexibility as an individual characteristic, defining it as an individual’s ability to manage conflicts using alternative and selectable approaches. This concept comprises three main elements: recognizing that effective choices are always available, being willing to adapt to situations (influenced by internal motivation), and having self-efficacy in choice behavior. In different learning environments and at different ages, an individual’s cognitive flexibility often varies and needs to be distinguished. For this, [27] ([27]) adapted the Cognitive Flexibility Inventory for use with Chinese high school students.

Cognitive flexibility is significantly positively correlated with communication skills. People with high cognitive flexibility are more confident and enthusiastic, and are more likely to succeed than those with low cognitive flexibility in communication ([20]). Cognitive flexibility is significantly positively correlated with confidence, self-efficacy, decisiveness, positive response and cooperation ([18]). In addition, individuals with high cognitive flexibility possess better social skills and positive social behaviors. Being argumentative is significantly positively correlated with cognitive flexibility, while verbal aggression is negatively correlated with cognitive flexibility ([19]). Cognitive flexibility is influenced by various factors such as emotional management, society and the environment ([27]). Individuals with poor cognitive flexibility are more likely to exhibit stubborn behaviors. This stubbornness will affect their acceptance of external information and interfere with their ability to make adjustments or adaptive responses. The application fields involved in cognitive flexibility are extremely extensive, such as education, psychological stress management, innovative thinking and other areas. In the learning process, the connection between cognitive flexibility and strategic flexibility (an individual’s response in the mathematical context) also warrants investigation ([12]).

### 1.4. The Present Study

This study aims to employ the “Four-Phase Flexibility Assessment” method developed by [37] ([37]) to examine the current state of strategic flexibility among students solving trigonometric identity transformation problems. It will analyze differences based on gender, class, etc., and explore the relationship between cognitive and strategic flexibility. The research findings are intended to offer suggestions for enhancing students’ strategic flexibility.

In particular, the present study is intended to address the following research questions:**Q1:** *How do high school students perform in terms of strategic flexibility when solving math problems?* The domain of these problems comes from the trigonometric identity transformation in mathematics.
**Q2:** *How do high school students differ in terms of strategic flexibility in external factors such as gender, class level, mathematics academic performance and subject combinations background?* The subject combinations background refers to the fact that in Chinese high school students, in addition to studying Chinese, mathematics and English, they also need to choose three courses from physics, chemistry, biology, history, geography and politics as their study subjects.
**Q3:** *What is the relationship between cognitive flexibility and strategic flexibility among high school students?*

Based on the aforementioned analysis, this study proposes the following hypotheses:

**H1:** 
*Strategic flexibility has two dimensions, and there is a certain connection between potential flexibility and actual flexibility.*


**H2:** 
*External factors can affect an individual’s potential flexibility and actual flexibility.*


**H3:** 
*Cognitive flexibility and strategic flexibility are correlated.*


## 2. Materials and Methods

### 2.1. Participants

This research recruited eleventh-grade students from a high school in Kunming Yunnan Province. The participants were spread across seven classes, with one advanced class and six standard classes. The school is considered second-tier in Yunnan Province’s high school admissions, attracting students with above-average academic achievement within the city. It should be noted that in a school in China, there is a unified plan and arrangement for the teaching content and teaching progress of major subjects (Chinese, Mathematics, English, etc.), which is called teachers’ collective lesson preparation. Before the formal class, teachers of the same subject in the same grade will carry out group study of textbooks and teaching materials according to the curriculum standards, formulate detailed teaching plans, determine specific teaching content and homework scope and other activities ([5]). Collective lesson preparation can enable students scattered in different classes to have a roughly the same learning process. In April 2024, researchers conducted an on-site survey, where students completed a test paper and a questionnaire. The researchers emphasized the study’s importance to the participants before beginning. At the time of the test, the students of this grade had just finished learning the relevant content of trigonometry. They can basically solve some conventional mathematical problems by applying the formulas of trigonometry. A total of 260 students participated in the survey, completing a test paper and a questionnaire. 237 (91.2%) test papers and questionnaires were considered valid. There are 102 boys and 135 girls.

### 2.2. Measures

#### 2.2.1. Test Paper of Strategic Flexibility

The initial draft of the test paper was constructed based on the content specifications for trigonometric identity transformation in the high school mathematics curriculum standards and related problems in the college entrance exam in China (Gaokao). The final test paper included three problems in total (See Appendix A for details). Each of the three problems in the test included multiple solutions. The style of test problems is quite different from the mathematical problems that students encounter in regular teaching. In regular teaching, mathematics problems emphasize the accuracy of students’ problem-solving. Generally, the solutions to problems are single, and the diversity of students’ problem-solving methods is not examined. The grading criteria and innovative strategies for each problem were collaboratively established through discussions between researchers and high school math teachers, ensuring that these test problems did not appear in regular teaching. Innovative strategies generally do not follow conventional problem-solving approaches. They have fewer steps and require less calculation, and need to handle problems tremendously flexibly based on the given conditions.

Students completed all three problems independently during a 45 min session in their regular math class. In addition to the widely used three stages of Three-Phase Flexibility Assessment, a fourth stage was incorporated. This additional stage required students to evaluate a list of strategies provided by researcher for each problem and identify the innovative solution. Data from this fourth stage served as a benchmark for potential flexibility. Consequently, the research process comprised four stages. Table 1 outlines the activities students engaged in during the testing phase.

The study consisted of four distinct phases. In the initial phase, participants were given three problems to solve independently. Each problem included a spacious area for students to document their approaches and answers. The researchers directed the students to solve all problems accurately and swiftly, showing all intermediate steps. If finished early, students were instructed to close their papers and wait quietly. This phase aimed to assess problem-solving accuracy and strategic flexibility.

The second phase required students to revisit the problems, starting from the first, and devise multiple solution strategies. An answer sheet with four response boxes was provided to record alternative approaches. Students were prohibited from modifying their initial responses and had to maintain the original problem order. This phase sought to evaluate students’ knowledge of diverse problem-solving techniques.

In the third phase, students reviewed their solutions from the previous stages and identified the innovative strategy they felt for each problem by marking it. No alterations to previous solutions were allowed. This phase aimed to determine if students could recognize innovative strategies. Upon completion, test papers were submitted to the examiners.

The final phase involved students receiving a scorecard listing various correct strategies for each problem, including standard solutions and the innovative ones. These strategies were presented in random order. Students were tasked with selecting the innovative strategy for each problem from the provided list. The researcher carefully balanced the number of lines and steps for each listed strategy to prevent students from identifying flexible strategies based solely on brevity. This phase shared the same objective as the third stage.

#### 2.2.2. Test Paper Coding

Three independent graduate students specializing in mathematics education evaluated student responses. Each dimension of coding was examined by at least two coders, with a third graduate student consulted to resolve any disagreements. The study’s primary focus is on assessing two types of flexibility: potential and actual. Actual flexibility is defined operationally as a learner’s ability to implement a flexible strategy for solving trigonometric identity transformation problems on their initial attempt. High actual flexibility is demonstrated by successfully using such a strategy, while low actual flexibility is indicated by relying solely on standard methods. This concept refers to the spontaneous application of strategy knowledge. Potential flexibility, on the other hand, is operationally defined as a learner’s knowledge of both standard and innovative strategies for a given problem. High potential flexibility is attributed to those who can generate multiple strategies, while low potential flexibility is assigned to those unable to do so. The scoring process for both types of flexibility requires the coding of the following constructs:Strategy Generation

The Strategy Generation score indicated the extent to which students knew multiple problem-solving solutions. For each question, researchers developed a coding system to assess students’ knowledge of standard and innovative solution methods. Evaluators examined responses from both initial and secondary stages of each problem. Students’ solutions were classified into three categories: standard, innovative, or other. In this evaluation, calculation mistakes were overlooked. All tentative solutions are encoded as “others”, including solutions in the form of guessing results, numerical verification, as well as incomplete, unclear and mathematical solutions that violate mathematical principles. Students who wrote both a standard and an innovative solution received a score of 1 for strategy generation on that particular problem. If not, their “Strategy Generation” score was zero. Presuming each student is scored for strategy generation on every problem, the highest possible score for strategy generation is 3 points.

2.Strategy Evaluation

The Strategy Evaluation assessed the students’ ability to pinpoint innovative solutions for each problem from their generated methods. Evaluators examined the strategies proposed by each student for every problem (from the initial to the second phase) and determined one or more flexible strategies. Subsequently, for each problem, students received 1 point if they selected a strategy in the third phase that evaluators deemed innovative. No points were awarded for choosing non-flexible solutions. The highest possible score for strategy evaluation is 3 points.

3.Potential Flexibility

The composite measure known as potential flexibility indicates a student’s awareness of multiple strategies and ability to identify an innovative solution among those known. This score is calculated for each problem based on two components: strategy generation and strategy evaluation. A student receives 1 point for potential flexibility on a given problem if they score 1 point for both strategy generation (showing both the standard strategy and the innovative strategy on the test paper) and strategy evaluation (correctly identifying the innovative strategy). If either component scores zero, the potential flexibility score for that problem is also zero. The highest achievable score for potential flexibility is 3 points.

4.Actual Flexibility

The concept of actual flexibility evaluates a student’s ability to employ a flexible approach in their initial problem-solving attempt. Some researchers also refer to this concept as practical flexibility ([37]). For each question, evaluators examined the student’s first solution strategy (during the initial phase) to assess its flexibility. Students received a score of 1 for actual flexibility on a particular problem if their first approach demonstrated flexibility; otherwise, they scored zero. The highest possible score for actual flexibility is 3 points.

5.Accuracy Score

Besides evaluating potential and actual flexibility, the study also assessed the correctness of each participant’s responses. The accuracy metric determined whether students could solve the provided equations correctly (obtaining the right numerical result) on their initial try. A score of 1 point was assigned for each correct answer, while incorrect responses received zero points. The highest possible accuracy score is 3 points.

6.Strategy Identification

To evaluate potential flexibility, researchers also coded strategy identification in the fourth stage, where participants selected from a list of provided solutions. This score indicated whether students could identify the innovative solution for each problem among the given options. Students earned 1 point for each problem where they chose the strategy deemed innovative by the evaluators. No points were awarded if a different strategy was selected. The highest possible score for strategy identification was 3 points.

#### 2.2.3. Cognitive Flexibility Questionnaire

This research utilized the revised “High School Student Cognitive Flexibility Questionnaire” developed by [27] ([27]). The value of Cronbach’s Alpha for total scale was 0.812, the test—retest reliability was 0.825, and the confirmatory factor analysis indicated that the three-factor model had better fitting indices. The instrument comprises 12 items rated on a 6-point scale (See Appendix B for details), assessing three dimensions: flexibility in choice, flexibility in willingness, and flexibility in efficacy. These dimensions are structured as follows:

Flexibility in Choice: Evaluates the awareness of making effective decisions in various situations. It encompasses items 1, 2, 3, 5, 6, and 10. Higher scores indicate greater awareness of flexible choice-making.

Flexibility in Willingness: Measures the readiness to adapt flexibly to different environments. It includes items 4, 11, and 12. Higher scores reflect increased flexibility in willingness.

Flexibility in Efficacy: Assesses self-efficacy in action selection. It consists of items 7, 8, and 9. Higher scores signify stronger flexibility in efficacy.

The questionnaire is self-assessment type, with participants responding based on personal experiences under standardized instructions. Students rate each statement’s applicability on a scale from 1 (“strongly disagree”) to 6 (“strongly agree”), completing the survey independently within five minutes after Four-Phase Flexibility Assessment. The 12 items use a 6-level rating scale: 1 (strongly disagree), 2 (disagree), 3 (somewhat disagree), 4 (somewhat agree), 5 (agree), and 6 (strongly agree). Four items (2, 3, 5, and 10) are reverse-scored. Total scores range from 12 to 72, with higher scores indicating greater cognitive flexibility.

## 3. Results

### 3.1. Test Paper Results Analysis

#### 3.1.1. Descriptive Statistics of Test Variables

Table 2 displays the overall average and standard deviation for all variables. Students showed a high level of proficiency in solving trigonometric identity transformation problems, with a mean score of 2.37, equivalent to a 79% success rate. This indicates that most participants were able to correctly answer the majority of the questions. The complexity of the three test problems increased sequentially, with correct answer rates of 93.67%, 81.43%, and 61.60%, respectively. The average score for strategy identification was 1.77, suggesting that approximately two-thirds of the students shared a similar understanding of innovative solutions. However, students exhibited low levels of strategic flexibility, with the mean score for actual flexibility being only 1.05. Potential flexibility was particularly low, with an average score of just 0.36.

#### 3.1.2. Correlation Between Actual Flexibility and Potential Flexibility

The descriptive statistics revealed low levels for both actual flexibility and potential flexibility, the two dimensions of mathematical thinking flexibility examined in this research. The score rate for actual flexibility exceeded that of potential flexibility. To further investigate their relationship, a paired sample T-test was employed. The results (Table 3) demonstrated that actual flexibility scores were significantly higher than potential flexibility scores (t = 11.898, *p* < 0.001). The paired sample correlation coefficient was 0.325, *p* < 0.001 (Table 4), suggesting a positive correlation between potential and actual flexibility, rather than independence.

#### 3.1.3. Analysis of Gender Differences

Researchers have shown significant interest in gender-based differences among learners as one of several non-intellectual factors influencing strategic flexibility. This research employed quantitative analysis to examine potential and actual thinking flexibility disparities between genders. The study designated gender as a characteristic variable, assigning “1” to males and “2” to females. An independent samples T-test was performed using the potential and actual flexibility scores from the test as variables (Table 5). For potential flexibility, Levene’s test for variance equality yielded an F-value of 1.401 (*p* = 0.238 > 0.1), indicating homogeneous variances. The subsequent test results (*p* = 0.407 > 0.05), based on equal variances, revealed no significant gender-based differences in potential cognitive flexibility. Likewise, for actual flexibility, Levene’s test produced an F-value of 0.378 (*p* = 0.539 > 0.1), again suggesting equal variances. The adjusted T-test outcome (*p* = 0.665 > 0.05) also showed no substantial gender-related differences in actual cognitive flexibility.

#### 3.1.4. Analysis of Class Level Differences

The study encompassed seven classes, with one designated as advanced and six as regular. Variations in instructional approaches and learning environments may have existed between these class categories. To assess strategic flexibility across different class levels, researchers employed an independent samples T-test, assigning a code of “1” to the advanced class and “0” to regular classes. Actual and potential flexibility scores served as the test variables. Results (Table 6) indicated that the advanced class had a marginally higher mean score for actual flexibility compared to regular classes, while potential flexibility mean scores were approximately equivalent. The two-tailed *p*-values for mean score differences in potential and actual flexibility were 0.959 and 0.224, respectively. With *p*-values exceeding 0.05, no statistically significant disparities in strategic flexibility scores were observed between students in advanced and regular classes.

#### 3.1.5. Correlation Between Strategic Flexibility and Mathematics Academic Performance

Developing flexibility in mathematical thinking may improve students’ capacity to adaptively apply knowledge and discover optimal solutions, potentially enhancing their problem-solving abilities and mathematics performance. This concept led to the hypothesis that a relationship exists between students’ potential/actual flexibility levels and their regular performance in mathematics. To investigate this proposed connection, researchers calculated Pearson correlation coefficients, using students’ mid-term math exam test scores as the measurement. The findings of this analysis are displayed in Table 7.

Analysis of the test outcomes revealed significant correlations between cognitive flexibility and mathematics academic performance. The Pearson correlation coefficients for potential and actual flexibility in relation to performance were found to be 0.302 and 0.259, respectively. Both correlations were statistically significant (*p* < 0.001). These findings demonstrate that both potential and actual cognitive flexibility, specifically in the context of trigonometric identity transformation problems, have a significant relationship with students’ mathematical academic performance at the 0.001 level, using a two-tailed test.

#### 3.1.6. Correlation Between Potential/Actual Flexibility and Subject Combinations

Yunnan Province introduced the “3 + 2 + 1” subject selection system for incoming high school students in 2022 as part of the fifth round of educational reforms. Among the seven classes studied, three subject groupings were identified: PCB (Physics-Chemistry-Biology), PCG (Physics-Chemistry-Geography), and PHG (Politics-History-Geography). Students in Grade 11 Class 7 opted for PHG, those in Grade 11 and Class 6 selected PCG, while the remaining classes chose PCB. To examine differences in strategic flexibility across subject combinations, researchers conducted a one-way ANOVA. As shown in Table 8, the analysis revealed significant difference in actual flexibility between the traditional science (PCB) and humanities (PHG) tracks (*p* = 0.005 < 0.05). Students who selected the PCB combination demonstrated higher levels of actual flexibility compared to those who chose PHG. However, no significant differences in actual flexibility were observed among other combinations, nor were there significant differences in potential flexibility.

### 3.2. Questionnaire Results Analysis

#### 3.2.1. Reliability and Validity Analysis of the Questionnaire

The survey instrument was structured into three sections, and the Cronbach’s Alpha values of each part are acceptable, as shown in Table 9. These findings indicate that the questionnaire exhibits strong internal consistency and is deemed suitable for research applications.

According to Table 10, the Kaiser-Meyer-Olkin (KMO) measure exceeded 0.7 for the entire questionnaire, indicating that the data is appropriate for factor analysis. Additionally, Bartlett’s Test of Sphericity yields significance values (Sig.) below 0.05, demonstrating high correlation among the items and confirming the questionnaire items’ validity.

#### 3.2.2. Descriptive Statistics of the Questionnaire Variables

An analysis of descriptive statistics was performed for the three dimensions of cognitive flexibility and their aggregate scores, as illustrated in Table 11. The overall mean score for cognitive flexibility was 43.44 (out of a possible 72), suggesting a moderate level of cognitive flexibility among the evaluated students.

In the Flexibility in Choice dimension, which is measured by items 1, 2, 3, 5, 6, and 10 (with higher scores signifying greater awareness of flexibility in choice), the mean score was 21.17 out of 36. This score, representing roughly two-thirds of the total possible points, indicates a slightly above-average awareness of flexibility in choice.

For the Flexibility in Willingness dimension, assessed by items 4, 11, and 12 (where higher scores denote greater willingness to be flexible), the mean score was 11.48 out of 18. This result reflects a moderate level of flexibility willingness among the students.

The Flexibility in Efficacy dimension, evaluated through items 7, 8, and 9 (with higher scores indicating greater efficacy in flexible behavior), yielded a mean score of 10.79 out of 18. This score, approximately two-thirds of the total possible points, suggests a moderate level of efficacy in flexibility.

Collectively, these three dimensions point to a moderate, slightly above-average overall cognitive flexibility among the tested student population.

#### 3.2.3. Correlation Between Cognitive Flexibility and Strategic Flexibility

To investigate the connection between cognitive flexibility and strategic flexibility, researchers employed Pearson correlation analysis. This analysis examined the relationship between cognitive flexibility scores and both potential and actual flexibility, as illustrated in Table 12. The results showed *p*-values exceeding 0.05 for both paired sample tests, suggesting no statistically significant associations between the variables in the paired test. Additionally, the study explored correlations between the three components of cognitive flexibility and both potential and actual flexibility. The findings revealed notable correlations at the 0.05 level (two-tailed) linking Flexibility in Willingness and Flexibility in Efficacy with both potential and actual flexibility.

## 4. Discussion

### 4.1. Fostering Students’ Willingness and Ability to Solve Problems with Multiple Solutions

Prior studies suggest that mathematical strategic flexibility is linked to various cognitive factors, including age, overall math proficiency, specific domain knowledge, executive functions, gender, and, to some degree, emotional aspects (e.g., [13]; [35]). While theoretical grounds exist for anticipating connections with these variables, findings have been inconsistent, except for the role of specific domain knowledge. This investigation revealed that students generally lack the inclination to solve problems using multiple methods, with some only exploring diverse approaches in their areas of expertise. This trend reflects both willingness and ability to solve problems with multiple solutions. The hesitation or subpar performance in multi-solution strategies may be partially attributed to emotional factors, as indicated by the cognitive flexibility survey results. Educators should help students recognize the benefits of multiple solutions in expanding mathematical understanding, enhancing cognitive breadth and depth, and fostering diligence ([2]; [4]; [15]).

However, mandating that all students engage in multi-solution problem-solving might be excessive. The implementation of multiple solutions should take into account students’ knowledge base and task complexity ([31]). To address this, teachers could promote the exchange and comparison of different problem-solving techniques among students. Building on their current problem-solving habits, peer discussions of various solutions can encourage active participation in collaborative learning and method comparison. Over time, teachers can increase the challenge by intentionally introducing solutions students may not have considered or presenting slightly flawed methods for analysis and comparison. Ultimately, requirements can be customized based on individual abilities, encouraging those with strong mathematical abilities to extensively explore multiple solutions and commending particularly innovative approaches ([34]). By encouraging students to share and present innovative solutions, the potential flexibility level of students is imperceptibly improved, thereby helping students to exert actual flexibility in real situations. For students with weaker mathematical performances, the priority should be on achieving correct answers before selecting the most innovative method from various solutions to refine their existing solutions ([23]; [35]).

### 4.2. Establishing Awareness of Strategy Optimization Through Challenging Problems

The results of test paper revealed a strong link between students selecting the innovative solution and their accuracy score, highlighting the significance of innovative approaches in mathematical problem-solving. Establishing awareness of strategy optimization can greatly enhance mathematical agility, boost problem-solving efficiency, and promote the growth of flexible thinking. The research also indicated that when confronted with the third, more intricate problem requiring more complex traditional methods, students exhibited greater actual flexibility compared to the first two problems. This implies that challenging questions are better at stimulating students’ motivation to seek flexible methods. Due to the characteristics of the subject, mathematics can provide students with opportunities to exercise strategic flexibility ([6]; [21]). Consequently, in problem-solving activities, teachers should frequently arrange some relatively difficult problems, which can significantly promote the development of strategic flexibility. It is suitable for boys, girls and students of different classes. In line with competency-oriented curriculum standards and college entrance examination evaluation system in China, the college entrance exam has introduced various innovative question types. Among these, ill-structured problems, often involving trigonometric identity transformations, form a crucial category. These problems typically require a higher level of thinking than structurally simple ones. As such, ill-structured problems provide an excellent opportunity for educators to nurture and improve students’ strategic flexibility. Teachers should effectively utilize resources like textbooks and college entrance exam questions, emphasizing the application and adaptation of structurally complex problems ([26]; [38]). In addition, science courses (physics, chemistry, etc.) in high school also have a certain relevance to flexibility. In the cultivation of flexibility, it is important to emphasize the fusion of subjects and scientific literacy ([16]; [17]). For instance, students who selected the PHG combination should enhance their studies in science courses. This also coincides with the current trend of countries vigorously developing scientific and engineering talents.

### 4.3. Emphasizing Variation, Transformation, and Connection in Classroom

Adopt educational approaches centered on variation, transformation, and connection. In terms of cognition, teachers need to make students aware of the existence of flexibility and have the idea that they can make adjustments or changes, and at the same time, teachers need to develop their belief that they can make changes successfully. Utilizing variation-based activities in teaching can encourage students to examine the fundamental principles, attributes, and distinctions of mathematical concepts or problems, enhancing their comprehension. These activities can also guide students to tackle mathematical tasks from fresh angles or predict outcomes, nurturing their abilities in transformation and connection. This method can also broaden students’ learning, fostering a problem-solving mindset that evolves from specific to general. Employing connection-focused strategies prompts students to clearly associate various aspects of concepts and consider problem directions holistically. Educators should thoroughly explore the subject matter, elucidate internal logic, grasp the core textbook structure, and guide students systematically and effectively, thus creating an environment conducive to developing students’ thinking skills. This subtle influence during the learning process can greatly improve students’ mathematical thinking flexibility. Carefully curated exercises and the implementation of diverse, multi-solution teaching activities are crucial. Developing students’ habits of finding multiple solutions to a single problem depends on teachers’ guidance and demonstration. In their regular instructional activities, teachers should focus on accumulating innovative, standardized, and scientifically sound questions. By selecting appropriate times for specialized problem-solving instruction, guiding students to explore multiple solutions, and offering valuable variant problems, teachers can facilitate the development of students’ abilities to connect and transform knowledge during problem-solving.

### 4.4. Limitations and Future Directions

This research has several limitations. First, the notion of flexibility is often intuitively grasped but difficult to express clearly, with attributes that can be experienced yet are challenging to describe precisely. In this investigation, strategic flexibility was categorized into potential and actual flexibility, evaluated through a four-phase assessment method validated by previous studies. However, judging and characterizing students’ strategic flexibility solely on the basis of test papers and survey questionnaire outcomes may be incomplete. Future research could explore alternative methods for assessing and grading strategic flexibility. Second, the study concentrated on trigonometric identity transformations, a subset of trigonometric functions that are more intricate and interrelated with other content. The complexity of these transformations is significantly higher compared to existing measures of strategic flexibility. Given the multiple steps in the tests and the time constraints, this intricacy could affect students’ performance and survey results. Third, the survey was limited to eleventh-grade students at one school in Kunming. This limitation hinders the generalizability of research results. Lastly, the research did not examine the impact of other individual characteristics and teaching methods in different regions, except for cognitive flexibility on strategic flexibility. In future studies, it would be advantageous to gather and verify data from multiple sources, considering students’ flexibility performance in relation to their personal characteristics and the different types of mathematical teaching approaches. This comprehensive strategy could then offer effective suggestions for instructional interventions aimed at improving strategic flexibility.

## 5. Conclusions

Eleventh-grade students demonstrated considerable accuracy in tackling trigonometric identity transformation problems. Nevertheless, both potential and actual flexibility were observed to be at low level, with a significant correlation between the two. Notably, actual flexibility surpassed potential flexibility. The underlying reasons merit further investigation. These outcomes support our conjecture that these flexibility types are separate yet interconnected, consistent with the findings of many psychological studies in the field of flexibility.

In terms of external factors, Examination of gender disparities showed no substantial differences between male and female students regarding potential and actual strategic flexibility. Likewise, an assessment of class distinctions revealed no significant variations in mathematical thinking flexibility between students in regular and advanced classes. However, the results also revealed significant correlations between strategic flexibility and mathematics academic performance. And students who selected the PCB combination demonstrated higher levels of actual flexibility compared to those who chose PHG. The learning of mathematics and science courses has a certain impact on students’ strategic flexibility.

The cognitive flexibility questionnaire results indicated above-average performance among students, particularly in areas of flexibility in choice, willingness, and efficacy. A positive association was found between cognitive flexibility and both potential and actual flexibility, implying that personality characteristics are crucial emotional factors affecting strategic flexibility.

## Figures and Tables

**Table 1 behavsci-15-01440-t001:** Testing Stages and Students’ Activities (time unit: minute).

Stage	Duration	Activity
Stage 1	15	Solve the three problems as quickly and accurately as possible.
Stage 2	15	Write down other solutions for the three problems.
Stage 3	5	Select the innovative solution from their own solutions for each problem.
Stage 4	5	Choose the innovative solution from the solutions provided by the testers for each problem.

**Table 2 behavsci-15-01440-t002:** Descriptive Statistics of Variables.

Variable	Mean	Standard Deviation
Accuracy	2.37	0.800
Strategy Generation	0.44	0.798
Strategy Evaluation	1.19	0.949
Strategy Identification	1.77	0.957
Actual Flexibility	1.05	0.825
Potential Flexibility	0.36	0.697

**Table 3 behavsci-15-01440-t003:** Paired Sample Test.

	Mean Difference	Standard Deviation	t	Degrees of Freedom	Sig. (Two-Tailed)
Potential Flexibility—Actual Flexibility	0.688	0.890	11.898	236	0.000

**Table 4 behavsci-15-01440-t004:** Paired Sample Correlation.

	Number of Cases	Correlation	Significance
Potential Flexibility— Actual Flexibility	237	0.325	0.000

**Table 5 behavsci-15-01440-t005:** Independent Samples Test for Gender Differences.

	Gender	Number of Cases	Mean	Standard Deviation	t	Sig. (Two-Tailed)
Potential Flexibility	Male	102	0.40	0.721	0.831	0.407
	Female	135	0.33	0.678		
Actual Flexibility	Male	102	1.02	0.820	−0.434	0.665
	Female	135	1.07	0.830		

**Table 6 behavsci-15-01440-t006:** Independent Samples Test for Class Level Differences.

	Class	Number of Cases	Mean	Standard Deviation	t	Sig. (Two-Tailed)
Potential Flexibility	Advanced Class	34	0.35	0.544	−0.052	0.959
	Regular Class	203	0.36	0.720		
Actual Flexibility	Advanced Class	34	1.21	0.845	1.220	0.224
	Regular Class	203	1.02	0.820		

**Table 7 behavsci-15-01440-t007:** Correlation Test between Mathematics Academic Performance and Strategic Flexibility.

	Number of Cases	Pearson Correlation	Sig. (Two-Tailed)
Potential Flexibility—Performance	237	0.302 ***	0.000
Actual Flexibility—Performance	237	0.259 ***	0.000

Note. *** indicates *p* < 0.001.

**Table 8 behavsci-15-01440-t008:** One-way ANOVA of Actual Flexibility/Potential Flexibility and Subject Combinations.

	Subject Combination	Subject Combination	Mean Difference	Standard Error	Sig. (Two-Tailed)
Actual Flexibility	PCB	PCG	0.099	0.154	1.000
		PHG	0.482 *	0.152	0.005
	PCG	PCB	−0.099	0.154	1.000
		PHG	0.383	0.198	0.163
	PHG	PCB	−0.482 *	0.152	0.005
		PCG	−0.383	0.198	0.163
Potential Flexibility	PCB	PCG	0.236	0.107	0.091
		PHG	0.182	0.132	0.430
	PCG	PCB	−0.236	0.107	0.091
		PHG	−0.053	0.151	0.979
	PHG	PCB	−0.182	0.132	0.430
		PCG	0.053	0.151	0.979

Note. * indicates *p* < 0.05.

**Table 9 behavsci-15-01440-t009:** Reliability Test for the Three Dimensions and Overall Cognitive Flexibility.

Dimension	Cronbach’s Alpha	Number of Items
Flexibility in Choice	0.664	6
Flexibility in Willingness	0.689	3
Flexibility in Efficacy	0.757	3
Cognitive Flexibility	0.691	12

**Table 10 behavsci-15-01440-t010:** Validity Test for the Three Dimensions and Overall Cognitive Flexibility.

Dimension	KMO Measure of Sampling Adequacy	Approximate Chi-Square	Significance	Degrees of Freedom
Flexibility in Choice	0.695	223.456	0.000	15
Flexibility in Willingness	0.636	127.581	0.000	3
Flexibility in Efficacy	0.668	181.628	0.000	3
Cognitive Flexibility	0.869	974.121	0.000	66

**Table 11 behavsci-15-01440-t011:** Descriptive Statistics of the Questionnaire Variables.

Variable	Mean	Standard Deviation
Flexibility in Choice	21.17	3.808
Flexibility in Willingness	11.48	2.913
Flexibility in Efficacy	10.79	2.966
Cognitive Flexibility	43.44	7.207

**Table 12 behavsci-15-01440-t012:** Paired Sample Correlations.

	Number of Cases	Correlation	Significance
Cognitive Flexibility—Potential Flexibility	237	0.121	0.063
Cognitive Flexibility—Actual Flexibility	237	0.064	0.323
Flexibility in Choice—Potential Flexibility	237	−0.007	0.909
Flexibility in Choice—Actual Flexibility	237	−0.111	0.088
Flexibility in Willingness—Potential Flexibility	237	0.176	0.007
Flexibility in Willingness—Actual Flexibility	237	0.159	0.014
Flexibility in Efficacy—Potential Flexibility	237	0.143	0.028
Flexibility in Efficacy—Actual Flexibility	237	0.153	0.018

## Data Availability

The raw data of this article will be made available by the authors, without undue reservation.

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
