# Peer review of "The Performance of Learners’ Strategic Flexibility and Its Relationship with External Factors and Cognitive Flexibility: A Survey of High School Mathematics in China"

_behavsci, 2025, doi:10.3390/bs15111440_

Round 1
Reviewer 1 Report
Comments and Suggestions for Authors
Review of the manuscript The Performance of Learners' Strategic Flexibility and Its Relationship with External Factors and Cognitive Aspects: A Survey of High School Mathematics in China
Thank you for an interesting and well-written article. You have made a great job in planning, conducting and reporting from this study. I have only a few minor remarks to make.
First, you have conducted your study in the context of trigonometry. It is an are of mathematics, where mastery of the subject relies heavily on the ability of using various trigonometric formula. On the one hand, for those students who can use these formula fluently, trigonometric tasks may appear as challenges that do not require extraordinary creativity or flexibility. On the other hand, for those students, who do not yet master these formula, the whole subject may be so challenging that they have no tools to demonstrate their creativity or flexibility. As you have collected your data from the grade 11 students, i.e., rather young learners, it might be fair to indicate already in the title and Introduction in which context this study has been conducted. I mean, this issue has much to do with the generalizability of your findings.
Second, I am not sure whether I understood correctly what is the origin of the framework of 2.2.2. Test Paper Coding, but if it is the study by Xu et al. (2017), should you then speak of potential and practical flexibility instead of potential and actual flexibility? Or what is the difference between actual and practical flexibility? In a way, I like your terminology more, because it emphasizes the momentary nature of flexibility, i.e., the it is dependent on the situation. Anyway, it might be helpful for the reader to clarify the difference between these notions if they are not the same.
Third, this is only a minor technical observation. In your tables, you usually give the means with two decimals and the other parameters with three decimals. I think that two decimals is enough for all, except for Sig-values, where it woud suffice to indicate whether p > .05, p < .05, p < 0.01, or p <.001.
Some more detailed issues:
* In Table 3, replace "Mean" with "Mean Difference".
* Table 8 is somewhat unclear to interpret. You say that you have performed Oneway ANOVA (i.e., analysis of mean differences) but the titles says that you report correlations. Moreover, you do so without giving any values of the relevant measures. Please, correct this.
* In Table 12, it would help reader if the included variables were ordered similarly as in the previous table, i.e.,
Flexibility in Choice - Actual Flexibility
Flexibility in Choice - Potential Flexibility
Flexibility in Willingness - Potential Flexibility
Flexibility in Willingness - Actual Flexibility
...
Author Response
Thank you very much for your pertinent comments, which are very helpful for us to improve the article.
The first comment you mentioned is about students' experience in using formulas and their background.
The students who participated in our experiment were 11th graders from the same school. They had just finished learning trigonometry at that time. Because Chinese schools have unified collective lesson preparation. Students of the same grade will basically receive largely the same teaching content. From the perspective of students, we believe that they are in the same teaching environment. The level they have demonstrated is exactly the flexibility we want to see in them. Of course, this is the situation of the investigated school. In the revised draft, we have made relevant supplements and descriptions of the participants' situations.
The second comment you pointed out is about the difference between actual flexibility and practical flexibility.
To be honest, "actual flexibility" and "practical flexibility" essentially mean the same thing in Chinese, but there are differences in their English expressions. However, in light of the purpose of this study, our authors unanimously agree that it is more appropriate to use "actual" to reflect that this is an individual's actual performance on the spot. In the revised draft, we added that some researchers defined this concept as practical flexibility.
In addition, the third comment you pointed out is about the data in the table. Thank you for pointing this out. We fully agree with this comment and have made corrections in the revised draft based on your feedback.
We look forward to your guidance and suggestions. Once again, we sincerely thank you for your comments!

Reviewer 2 Report
Comments and Suggestions for Authors
Thank you for the opportunity to review this interesting manuscript, which addresses an important educational issue—namely, the performance of learners’ strategic flexibility and its relationship with external factors and cognitive flexibility in math classes in Chinese high schools. In the file attached I have outlined several concerns and comments intended to support the improvement of the manuscript.

Author Response
Thank you very much for your comments, which help us a lot to improve the article.
The first and second comments you mentioned are about modifying the title and supplementing the research hypotheses.
In the revised draft, we have made relevant supplements and descriptions on this. At the same time, based on the third and fourth comments you pointed out (regarding the conclusion and limitation positions), we have also made adjustments to the relevant positions.
The fifth and sixth comments you pointed out are about supplementing “1.3. Cognitive Flexibility” and the reliability and validity of measurement tools.
In the revised draft, we added content on cognitive flexibility and reported on the reliability and validity of the scale. The scale is about the cognitive flexibility of high school students and is in line with our measurement subjects, so no additional adjustments were made.
The seventh comment you mentioned is in the discussion section.
In the revised draft, we also had certain discussions on this, adjusted the relevant statements and supplemented the references.
The eighth comment you mentioned is about the basic situation of the participants.
In the revised draft, we have made new additions to the backgrounds of the participants and their experiences during the tests. Although they are distributed in different classes, due to the collective lesson preparation by Chinese teachers, the participants can be regarded as having similar learning experiences.
The ninth comment you mentioned is about the expression of "demonstrating knowledge of both standard and innovation strategies".
The mention of "demonstrating knowledge of both standard and innovation strategies" here actually does not mean that students present certain knowledge, but rather that researchers evaluate students based on their solutions in the test papers. There was ambiguity in the expression before. In the revised draft, we have redescribed it. In addition, for the concept of strategy generation, we are based on the relevant definition given in Article (Xu et al., 2017). The Strategy Generation score indicated the extent to which students knew multiple problem-solving solutions.
Furthermore, the tenth comment you mentioned is about “unorthodox methods”. We have redescribed the relevant content. The previous expression was a bit unclear. We hope the new expression can explain the relevant issues clearly for you.
We look forward to your guidance and suggestions. Once again, we sincerely thank you for your comments!

Round 2
Reviewer 2 Report
Comments and Suggestions for Authors
Thank you for the opportunity to review the revised version of the manuscript. The following concerns and comments are provided to support the further development of the manuscript.
- The research question/s is/are still missing. The authors focused only on the comment regarding the research hypotheses.
- The cognitive flexibility section was enriched. However, the paragraph added is not bibliographically supported. Furthermore, it is debatable whether cognitive flexibility should be regarded as a personality trait. In fact, cognitive flexibility is not one of the "Big Five" personality traits. Martin et al. (1995, 1998) claim that “cognitive flexibility refers to a person’s awareness of communication alternatives, willingness to adapt to the situation, and self-efficacy in being flexible”. Thus, it is not classified under established personality trait models. The authors are encouraged to elaborate on this section further.
- Statements such as the following ones appear arbitrary if not supported by the literature:
- “Mathematics holds an important position in the cultivation of flexibility” (line 499),
- “science courses (physics, chemistry, etc.) in high school 511 also have a certain relevance to flexibility” (lines 511-512),
- “However, mandating that all students engage … from various solutions to 488 refine their existing solutions” (lines 474-489)
The above list is indicative. The authors are encouraged to carefully revise the manuscript in order to ensure that the arguments are more thoroughly supported by the existing literature.
Author Response
Thank you very much for your comments this time. We attach great importance to the relevant comments and issues you mentioned, as they have provided significant assistance for the improvement of the article.
The first comment you mentioned is that the research problems are missing.
Our article originally discussed the research aim. Based on the comment you mentioned, we clarified the research questions (1.4 the Present Study, lines 160-172).
The second comment you mentioned is about the lack of references in the cognitive flexibility part and whether cognitive flexibility should be regarded as a personality trait.
Based on your comment, we have refined the corresponding references (1.3 Cognitive Flexibility, lines 138, 140, 144, 145 and 152). Regarding the issue that cognitive flexibility does not fall under the personality trait model, the personality traits we refer to here actually belong to a broader category of individual characteristics. However, it is indeed easy to misinterpret cognitive flexibility as one of the "Big Five" personality traits. We fully respect your opinion and change the personality traits in the article to individual characteristics. Personality traits (lines 128, 151 and 572) appeared three times in the original text. We have deleted and modified them, changing the first and third to personality traits and deleting the second one (lines 128, 151 and 572).
Furthermore, according to the third comment you mentioned, some of the content is not supported by literature.
Based on your opinions, we have supplemented the corresponding references (4. discussion, lines 491, 500, and 504). In the discussion section, we mentioned that mathematics and science courses play a certain role in cultivating flexibility. The relevant statements mainly stem from the actual teaching practices of various subjects in high schools. Compared with other subjects in high school (such as literature, foreign languages, history, etc.), Many mathematical contents require students to think flexibly and solve problems flexibly in the process of teaching and learning. That is to say, the subject of mathematics offers students more opportunities to exercise strategic flexibility. These characteristics are not prominently reflected in other non-scientific subjects. Therefore, it appears rather important. Similarly, strategic flexibility is also required in science courses. Of course, we have also revised the relevant expressions and supplemented the relevant literature (lines 516-517, and 531).
Once again, we sincerely thank you for your review and the valuable comments you have provided!

Round 3
Reviewer 2 Report
Comments and Suggestions for Authors
In this revised version, the authors have carefully and satisfactorily addressed the concerns raised in the previous two rounds of review. The revisions have notably improved the clarity and overall quality of the manuscript. The authors have thoughtfully addressed all comments, and no major issues remain.
I therefore recommend acceptance of the manuscript pending minor revision to further strengthen it.
More precisely, the phrase “Due to the characteristics of the subject, mathematics can provide students with more opportunities to exercise strategic flexibility” could be refined by removing the word “more” (line 515).